# Cognitive emotion regulation questionnaire: Evidence of internal structure through confirmatory factor modeling and exploratory structural equation modeling

Pablo Ezequiel Flores-Kanter[1,2,3]* , Luciana Moretti[1], Zoilo Emilio García-Batista[4],
Leonardo Medrano[1,4]

1 Universidad Siglo 21, Córdoba, Argentina, 2 Universidad Santo Tomás, Villavicencio, Colombia,
3 Universidad Católica de Córdoba, Córdoba, Argentina, 4 Pontificia Universidad Católica Madre y
Maestra, Santiago de los Caballeros, República Dominicana

☯ These authors contributed equally to this work.
* pablo.floreskanter@ues21.edu.ar

## Abstract

The Cognitive Emotion Regulation Questionnaire is one of the most widely used instruments to measure cognitive emotion regulation, and its psychometric properties have been evaluated in various studies and cultural contexts. However, the 9-factor internal structure originally proposed for the scale measures does not present consistent evidence in the literature. The exclusive use of the confirmatory factor modeling in previous literature may largely explain this inconsistency. In the present research, we propose to estimate innovative measurement models for this questionnaire, the exploratory structural equation modeling. For this purpose, we worked with a large sample of Argentines of 6881 Argentine adults aged between 18 and 81 (M = 27.14, SD = 9.86; 69.6% female; 86.4% not being in psychological or psychiatric treatment) and compared the fit of the models. The results favored the exploratory structural equation model ($\chi 2 = 4092.89$, df = 342, CFI = .982, SRMR = .013, RMSEA = .04), indicating that we are not observing indicators that reflect simple factorial structures. Based on the results of the present study, it can be concluded that the simple 9-factor structure traditionally proposed for the CERQ does not exhibit adequate model fit. This finding suggests the need for caution among applied psychologists and clinical researchers who interpret scale scores as if they reflect distinct, unidimensional factors. The common practice of generating summed scores across items associated with each of the originally proposed factors may therefore be questionable. A more nuanced understanding of the questionnaire's internal structure may ultimately enhance the reliability and interpretability of CERQ scores in both clinical and applied settings, thereby improving the quality of psychological assessment and intervention outcomes.

**Data availability statement:** The data that support the findings of this study are openly available in Mendeley Data at http://doi.org/10.17632/48y8tkf5wh.4.

**Funding:** The author(s) received no specific funding for this work.

**Competing interests:** The authors have declared that no competing interests exist.

## Introduction

Understanding how individuals regulate their emotions is fundamental to explaining human behavior, mental health, and psychological resilience [1–3]. Emotional regulation is not only a core mechanism in everyday functioning but also a critical factor in the onset, maintenance, and treatment of numerous psychological disorders [4,5]. Given its centrality, the ability to accurately measure emotional regulation—particularly through cognitive strategies—has become a pressing concern in both clinical and research contexts [6]. Reliable and valid assessment tools are essential for identifying dysfunctional patterns, designing targeted interventions, and advancing theoretical models of emotion regulation.

One of the most widely used instruments for assessing cognitive emotion regulation strategies is the Cognitive Emotion Regulation Questionnaire (CERQ). Developed to evaluate how individuals cognitively manage negative or stressful life events [7], the CERQ captures a broad spectrum of regulatory strategies, including both adaptive (e.g., acceptance, positive reappraisal) and maladaptive (e.g., rumination, catastrophizing) responses. Since its introduction, the CERQ has been extensively applied in diverse populations and settings, serving as a reference point for both clinical diagnostics and empirical research. However, despite its widespread use, important psychometric limitations remain (particularly concerning its internal structure and the theoretical independence of its factors) raising critical questions about the validity of the interpretations of its scores and the assumptions underlying its factorial model.

A review of previous studies examining the psychometric properties of the CERQ indicates that its original nine-factor model often fails to achieve adequate fit without post hoc modifications. These include correlating residual errors [8–12], removing several items [13], or allowing items to load on theoretically distinct factors [9,11]. Moreover, there is consistent evidence of factor overlap, particularly between rumination and catastrophizing [4,11,13], which undermines the conceptual clarity and discriminant validity of these constructs. Additional signs of model misfit have also been reported, such as low (<.30), nonsignificant, or even negative factor loadings [9,13,14]. These issues are particularly salient in the short version of the CERQ, where certain items exhibit complex or ambiguous behavior and latent factors continue to overlap [15].

To date, the evaluation of the CERQ's internal structure has relied almost exclusively on Confirmatory Factor Analysis (CFA), predominantly using the maximum likelihood estimation method [8,12,14,16–18]. CFA models assume that each item loads solely on its designated factor, prohibiting cross-loadings. This implies that items are treated as pure indicators of their constructs—an assumption that often does not hold in psychological measurement [19]. In reality, items may share variance with multiple latent constructs, particularly when measuring related domains [20]. When such cross-loadings are ignored—i.e., constrained to zero—the unmodeled complexity is absorbed elsewhere in the model, leading to biased parameter estimates and reduced model validity [21]. For this reason, accounting for cross-loadings is essential, especially in scales like the CERQ that aim to assess conceptually interconnected strategies [22].

In light of these concerns, the present study aims to conduct a comprehensive evaluation of the internal structure of the CERQ using advanced factorial modeling techniques [23,24]. Specifically, we compare the traditional Confirmatory Factor Analysis (CFA) approach with Exploratory Structural Equation Modeling (ESEM), including a variant known as Set-ESEM (Fig 1), to determine which framework provides a more accurate and theoretically coherent representation of the scale [25]. We hypothesize that: (a) ESEM models will demonstrate superior parameter estimation compared to the CFA model, particularly in terms of standardized factor loadings and interfactor correlations; and (b) within the ESEM framework, the Set-ESEM model will yield a more parsimonious yet well-fitting solution by allowing cross-loadings within conceptually related strategy groups while restricting them across theoretically distinct domains.

## Methods

### Participants

Although a prior calculation of the required sample size was not conducted, a sufficiently large sample was anticipated to ensure both the stability (i.e., that the estimation algorithm would converge on an admissible solution) and the accuracy of the parameter estimates (i.e., that the estimated parameter values would not substantially differ from the true population values) derived from the tested models [26,27]. The simulation studies reported by Wolf et al. [26] suggest a minimum required sample size of 500 participants, considering the complexity of the models evaluated here (e.g., number of parameters to be estimated, number of items per factor). Here, the sample consisted of 6881 Argentine adults aged between 18 and 81 (M = 27.14, SD = 9.86). Of the total sample, 69.6% (n = 4,792) identified as female, while 86.4% (n = 5,948) reported not being in psychological or psychiatric treatment. All participants were adequately informed of the research objectives, the anonymity of their responses, and their voluntary participation. It was also made clear to them that participating would not cause them any harm and that they could leave the study whenever they wished. Informed consent was obtained from all participants by clicking the 'I Accept' button on the online form. Participants did not receive any specific incentive for participation. International ethical guidelines for human studies were included [28]. The ethics committee of the Research Secretariat of the Universidad Siglo 21 previously approved the research protocol following the ethical guidelines of the APA. For participant recruitment, an open-mode online sampling method was used [29]. This data collection methodology is equivalent to traditional forms of collection (i.e., face-to-face), yielding equality of means, internal consistencies, intercorrelations, response rates, and comfort level in completing the questionnaires [30]. It is important to note that, given the data collection method, there were no missing data, and the only inclusion criteria were agreeing to participate in the study and being 18 years of age or older. The sample for this observational, cross-sectional study was collected using an

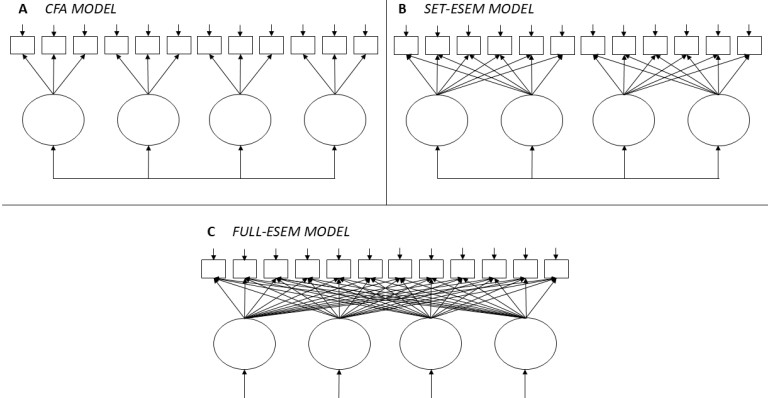

**Fig 1. Schematic comparison of CFA, Set-ESEM and Full-ESEM models.**

online survey format to collect information through the Google Forms platform and delivered via Facebook social media. The recruitment period was from April 15, 2019, to August 18, 2021.

### Instruments

Cognitive Emotion Regulation Questionnaire (CERQ). The CERQ is a self-report instrument composed of 36 items answered on a Likert-type scale, where 1 is almost never, and 5 is almost always [10]. This questionnaire assesses nine ER cognitive strategies commonly used in the face of aversive events: self-blame, blaming others, rumination, catastrophizing, putting into perspective, positive refocusing, positive reappraisal, acceptance and planning. The Argentine version of the scale was used [11]. This version was adapted and validated in a sample of university students (N = 359, M age = 24.6, female = 50.1). The model of 9 correlated factors showed acceptable fit indicators ($\chi2$ = 875.50, df = 538, CFI = .91, GFI = .90, RMSEA = .04), and the internal consistency indicators measured by Cronbach's Alpha varied between .59 and .83.

### Statistical analysis

To evaluate the internal structure of the Cognitive Emotion Regulation Questionnaire (CERQ), three models with increasing degrees of flexibility were estimated: [1] Confirmatory Factor Analysis (CFA) [2], Set-ESEM, and [3] Full-ESEM. This order reflects a progression from more restrictive to more flexible modeling approaches. CFA assumes that each item loads exclusively on its designated factor, with all cross-loadings fixed to zero. In contrast, Full-ESEM allows all items to load on all factors, while Set-ESEM offers an intermediate solution: it permits cross-loadings within predefined, theoretically coherent groups of factors, but constrains them across groups [25].

In our study, the nine cognitive emotion regulation strategies were divided into two theoretically grounded and functionally distinct groups. CERS Group 1—comprising rumination, catastrophizing, self-blame, and other-blame—includes maladaptive strategies characterized by their implicit, automatic, and reactive nature. These strategies tend to be activated with little cognitive control, often in response to stress or emotional threat, and have been associated with negative affect, perseverative thinking, and greater vulnerability to psychopathology. In contrast, CERS Group 2—comprising acceptance, putting into perspective, positive refocusing, positive reappraisal, and planning—includes more adaptive strategies that require deliberate, effortful processing and are typically mediated by executive functions such as inhibition, working memory, and cognitive flexibility. These strategies reflect top-down emotional regulation and are associated with greater psychological resilience, wellbeing, and functional coping. The distinction between these two groups is supported by dual-process models of emotion regulation, which differentiate between bottom-up, stimulus-driven responses and top-down, cognitively regulated mechanisms, as well as by neurocognitive evidence showing differential activation patterns in brain networks involved in automatic versus controlled regulation processes [4,31]. This grouping provided the theoretical basis for specifying the Set-ESEM model, allowing cross-loadings within each group to account for conceptual overlap, while constraining cross-loadings across groups to preserve parsimony and interpretability.

Models were analyzed using weighted mean squares with adjusted mean and variance (WLSMV) as the estimator, given the categorical nature of the observable variables [32,33]. In the case of the ESEM models, factors were rotated using Target and Geomin (Oblique) rotations [25]. To interpret the factorial solutions loadings of .40, .55 and .70 were considered low, medium and high, respectively [34]. Regarding the size of the factorial correlations, values of .10, .30 and .50 were considered small, medium and large, respectively [35].

The fit of the factor models was assessed with three complementary indices: the comparative fit index (CFI), the root mean square error of approximation (RMSEA) and the standardized root mean square residual (SRMR). It has been suggested that CFI values greater than or equal to 0.90 and 0.95 reflect an acceptable and excellent fit to the data, while RMSEA values less than 0.08 and 0.05 may indicate a reasonable and close fit to the data, respectively [19,36,37]. In the

case of SRMR, a value less than or equal to .08 has been found to indicate a good fit for the data [36,37]. However, since it is to be expected by the very specification of ESEM models that they achieve better fit indices [25], the most important criterion lies in the comparison of the parameters estimated by the CFA and ESEM models. According to Marsh et al. [25], ESEM models are most appropriate when the multiple ESEM factors are well defined in the measurement model, and there are substantively important differences in the parameter estimates based on the CFA and ESEM models (i.e., ESEM would typically show significant non-trivial cross-loadings, as well as different primary loadings and lower factor correlations than CFA, which would generally overestimate factor correlations due to omitted cross-loadings) [19,23]. In the latter case, the factor structure is considered complex, and ESEM would be the optimal model. If the ESEM models are sufficiently similar to the CFA results, this would support the simple structure, and indicate that the CFA is the optimal model. The aforementioned statistical analyses were performed with Mplus (version 8).

## Results

The first step consisted of verifying the fit of the CFA, Full-ESEM, and Set-ESEM models. In the Set-ESEM model with Target rotation, the standard errors of the estimated model parameters could not be computed because the model could not be identified. As for the rest of the estimated models, while the CFA model presents an acceptable fit to the data, the Full-ESEM and Set-ESEM models achieve a better fit (Table 1). Since this is to be expected given the models' specifications [25], the most crucial step lies in comparing the estimated parameters.

Second, the estimated parameters for these models were verified (except for the Set-ESEM model with Target rotation that could not be identified; see Tables 2 and 3). In the case of the CFA, high interfactor correlations are evident (≥.50; see Table 2). More specifically, between the latent factors of self-blame, rumination, and catastrophizing, the interfactor correlations obtained were .67, .57, .75; between catastrophizing and positive reinterpretation of −.51; between positive refocusing, putting in perspective, positive reinterpretation, and focus on plans of .55, .52, .67, .68, .65, .63, .54, .51, .81. When the correlation matrix of the CFA is compared with the model that follows it in restrictive terms (from more to less restrictive models), the Set-ESEM, it is verified that some correlations decrease in magnitude (Table 2). For example, among the latent factors self-blame, rumination, and catastrophizing, the interfactor correlations obtained were .38, .43, .48. The same is evident for the case of the interfactor correlations between positive refocusing, putting into perspective, positive reinterpretation, and focus on plans. Unexpectedly and contrary to the initial hypothesis, an increase in the magnitude of some of the interfactor correlations is also verified in the Set-ESEM model, specifically, between the latent factor of rumination and the latent factors of acceptance, putting in perspective and focusing on plans. In addition, theoretically, incongruent correlations were observed in the Set-ESEM model with oblique rotation (Table 2). These correspond to the interfactor correlations between the latent factor rumination and the latent factors' acceptance, putting in perspective, and focus on plans (.21, .21, .44).

Following in order of restriction, in the Full-ESEM models with target and oblique rotation, a similar decrease in the interfactor correlations already described is verified (Table 3). The Full-ESEM model with target rotation also shows a

**Table 1. Comparison of fit between CFA, Full-ESEM and set-ESEM models.**

| Model | $\chi^2$ | df | CFI | SRMR | RMSEA (90% CI) |
|---|---|---|---|---|---|
| CFA | 21007.56 | 558 | .900 | .057 | .073 (.072, .074) |
| Set-ESEM | 9140.25 | 462 | .959 | .026 | .052 (.051 ,.053) |
| Full-ESEM | 4092.89 | 342 | .982 | .013 | .040 (.039, .041) |

ESEM = exploratory structural equation modeling; CFA = confirmatory factor analysis; $\chi^2$ = chi-square; df = degrees of freedom; CFI = comparative fit index; SRMR = standardized root mean square residual; RMSEA = root mean square error of approximation; $p < .001$ for all chi-square tests of model fit. Since in ESEM models the fitting results do not vary as a function of the rotation method (Marsh et al., 2019), the table does not differentiate the fit as a function of these methods.

**Table 2. Standardized regression weights for the CFA and Set-ESEM models (Oblique rotation).**

| Ítem/Factor | CFA | | | | | | | | | Set-ESEM Oblique | | | | | | | | |
|---|---|---|---|---|---|---|---|---|---|---|---|---|---|---|---|---|---|---|
| | F1 | F2 | F3 | F4 | F5 | F6 | F7 | F8 | F9 | F1 | F2 | F3 | F4 | F5 | F6 | F7 | F8 | F9 |
| **S-B** | | | | | | | | | | | | | | | | | | |
| i01 | **.84** | .00 | .00 | .00 | .00 | .00 | .00 | .00 | .00 | **.69** | −.06 | .24 | −.04 | .00 | .00 | .00 | .00 | .00 |
| i17 | **.80** | .00 | .00 | .00 | .00 | .00 | .00 | .00 | .00 | **.92** | .05 | −.05 | .03 | .00 | .00 | .00 | .00 | .00 |
| i26 | **.63** | .00 | .00 | .00 | .00 | .00 | .00 | .00 | .00 | .24 | **.45** | .11 | −.01 | .00 | .00 | .00 | .00 | .00 |
| i33 | **.66** | .00 | .00 | .00 | .00 | .00 | .00 | .00 | .00 | **.52** | .26 | .02 | .00 | .00 | .00 | .00 | .00 | .00 |
| **Rum.** | | | | | | | | | | | | | | | | | | |
| i03 | .00 | **.60** | .00 | .00 | .00 | .00 | .00 | .00 | .00 | −.02 | **.45** | .25 | .00 | .00 | .00 | .00 | .00 | .00 |
| i15 | .00 | **.78** | .00 | .00 | .00 | .00 | .00 | .00 | .00 | .06 | .32 | **.48** | −.04 | .00 | .00 | .00 | .00 | .00 |
| i18 | .00 | **.51** | .00 | .00 | .00 | .00 | .00 | .00 | .00 | .04 | **.64** | −.01 | .06 | .00 | .00 | .00 | .00 | .00 |
| i27 | .00 | **.81** | .00 | .00 | .00 | .00 | .00 | .00 | .00 | −.02 | **.47** | **.46** | −.02 | .00 | .00 | .00 | .00 | .00 |
| **Catas.** | | | | | | | | | | | | | | | | | | |
| i08 | .00 | .00 | **.35** | .00 | .00 | .00 | .00 | .00 | .00 | .01 | .13 | .19 | .22 | .00 | .00 | .00 | .00 | .00 |
| i10 | .00 | .00 | **.87** | .00 | .00 | .00 | .00 | .00 | .00 | .07 | −.03 | **.81** | .04 | .00 | .00 | .00 | .00 | .00 |
| i22 | .00 | .00 | **.58** | .00 | .00 | .00 | .00 | .00 | .00 | −.05 | .02 | **.52** | .18 | .00 | .00 | .00 | .00 | .00 |
| i35 | .00 | .00 | **.86** | .00 | .00 | .00 | .00 | .00 | .00 | .02 | .09 | **.76** | .05 | .00 | .00 | .00 | .00 | .00 |
| **O-B** | | | | | | | | | | | | | | | | | | |
| i09 | .00 | .00 | .00 | **.79** | .00 | .00 | .00 | .00 | .00 | −.01 | −.06 | .04 | **.78** | .00 | .00 | .00 | .00 | .00 |
| i21 | .00 | .00 | .00 | **.59** | .00 | .00 | .00 | .00 | .00 | .00 | .27 | −.11 | **.60** | .00 | .00 | .00 | .00 | .00 |
| i29 | .00 | .00 | .00 | **.90** | .00 | .00 | .00 | .00 | .00 | .03 | −.04 | .01 | **.90** | .00 | .00 | .00 | .00 | .00 |
| i36 | .00 | .00 | .00 | **.79** | .00 | .00 | .00 | .00 | .00 | −.03 | .03 | .04 | **.76** | .00 | .00 | .00 | .00 | .00 |
| **Acept.** | | | | | | | | | | | | | | | | | | |
| i02 | .00 | .00 | .00 | .00 | **.53** | .00 | .00 | .00 | .00 | .00 | .00 | .00 | .00 | **.67** | −.07 | −.05 | .08 | −.05 |
| i16 | .00 | .00 | .00 | .00 | **.81** | .00 | .00 | .00 | .00 | .00 | .00 | .00 | .00 | **.73** | −.01 | .02 | .14 | −.01 |
| i32 | .00 | .00 | .00 | .00 | **.70** | .00 | .00 | .00 | .00 | .00 | .00 | .00 | .00 | **.60** | .04 | .09 | −.04 | .11 |
| i25 | .00 | .00 | .00 | .00 | −.21 | .00 | .00 | .00 | .00 | .00 | .00 | .00 | .00 | .39 | .22 | .02 | **−.75** | .04 |
| **Ref.** | | | | | | | | | | | | | | | | | | |
| i04 | .00 | .00 | .00 | .00 | .00 | **.80** | .00 | .00 | .00 | .00 | .00 | .00 | .00 | −.02 | **.66** | −.02 | .25 | −.05 |
| i14 | .00 | .00 | .00 | .00 | .00 | **.80** | .00 | .00 | .00 | .00 | .00 | .00 | .00 | .03 | **.86** | −.03 | −.06 | .04 |
| i24 | .00 | .00 | .00 | .00 | .00 | **.85** | .00 | .00 | .00 | .00 | .00 | .00 | .00 | .00 | **.85** | .05 | .03 | −.04 |
| i28 | .00 | .00 | .00 | .00 | .00 | **.87** | .00 | .00 | .00 | .00 | .00 | .00 | .00 | .00 | **.76** | .05 | .07 | .06 |
| **Pers.** | | | | | | | | | | | | | | | | | | |
| i07 | .00 | .00 | .00 | .00 | .00 | .00 | **.56** | .00 | .00 | .00 | .00 | .00 | .00 | .08 | .01 | **.48** | .02 | .01 |
| i11 | .00 | .00 | .00 | .00 | .00 | .00 | **.65** | .00 | .00 | .00 | .00 | .00 | .00 | .03 | −.03 | **.75** | −.04 | −.03 |
| i20 | .00 | .00 | .00 | .00 | .00 | .00 | **.81** | .00 | .00 | .00 | .00 | .00 | .00 | −.09 | .02 | **.68** | .19 | .02 |
| i34 | .00 | .00 | .00 | .00 | .00 | .00 | **.78** | .00 | .00 | .00 | .00 | .00 | .00 | .10 | .01 | **.75** | .00 | −.02 |
| **Reint.** | | | | | | | | | | | | | | | | | | |
| i06 | .00 | .00 | .00 | .00 | .00 | .00 | .00 | **.77** | .00 | .00 | .00 | .00 | .00 | .14 | .01 | .07 | **.52** | .20 |
| i12 | .00 | .00 | .00 | .00 | .00 | .00 | .00 | **.82** | .00 | .00 | .00 | .00 | .00 | .15 | .06 | .13 | **.48** | .20 |
| i23 | .00 | .00 | .00 | .00 | .00 | .00 | .00 | **.80** | .00 | .00 | .00 | .00 | .00 | −.01 | .03 | .33 | **.59** | .04 |
| i31 | .00 | .00 | .00 | .00 | .00 | .00 | .00 | **.91** | .00 | .00 | .00 | .00 | .00 | .02 | .16 | .25 | **.61** | .06 |
| **F-P** | | | | | | | | | | | | | | | | | | |
| i05 | .00 | .00 | .00 | .00 | .00 | .00 | .00 | .00 | **.74** | .00 | .00 | .00 | .00 | .07 | .02 | −.05 | .35 | **.42** |
| i13 | .00 | .00 | .00 | .00 | .00 | .00 | .00 | .00 | **.88** | .00 | .00 | .00 | .00 | .05 | .05 | −.03 | .38 | **.51** |
| i19 | .00 | .00 | .00 | .00 | .00 | .00 | .00 | .00 | **.58** | .00 | .00 | .00 | .00 | −.05 | −.14 | .08 | −.01 | **.81** |
| i30 | .00 | .00 | .00 | .00 | .00 | .00 | .00 | .00 | **.75** | .00 | .00 | .00 | .00 | .00 | .03 | −.01 | .21 | **.62** |

*(Continued)*

**Table 2.** (Continued)

| Ítem/Factor | CFA | | | | | | | | | Set-ESEM Oblique | | | | | | | | |
|---|---|---|---|---|---|---|---|---|---|---|---|---|---|---|---|---|---|---|
| | F1 | F2 | F3 | F4 | F5 | F6 | F7 | F8 | F9 | F1 | F2 | F3 | F4 | F5 | F6 | F7 | F8 | F9 |
| **Interfactor Correlation** | F1 | F2 | F3 | F4 | F5 | F6 | F7 | F8 | F9 | F1 | F2 | F3 | F4 | F5 | F6 | F7 | F8 | F9 |
| F1 | 1.00 | | | | | | | | | 1.00 | | | | | | | | |
| F2 | .67 | 1.00 | | | | | | | | **.38** | 1.00 | | | | | | | |
| F3 | .57 | .75 | 1.00 | | | | | | | .43 | .48 | 1.00 | | | | | | |
| F4 | −.02 | .27 | .42 | 1.00 | | | | | | −.10 | **.07** | .34 | 1.00 | | | | | |
| F5 | −.04 | −.02 | −.27 | −.11 | 1.00 | | | | | −.00 | *.21* | **−.16** | −.06 | 1.00 | | | | |
| F6 | −.32 | −.32 | −.40 | .00 | .44 | 1.00 | | | | −.29 | **−.13** | −.40 | .04 | .35 | 1.00 | | | |
| F7 | −.07 | −.06 | −.25 | .04 | .55 | .52 | 1.00 | | | −.05 | *.21* | −.26 | .06 | **.44** | .46 | 1.00 | | |
| F8 | −.27 | −.25 | −.51 | −.11 | .67 | .68 | .65 | 1.00 | | −.31 | **−.12** | **−.16** | −.06 | **.47** | **.52** | **.37** | 1.00 | |
| F9 | −.13 | .02 | −.33 | −.03 | .63 | .54 | .51 | .81 | 1.00 | −.05 | *.44* | −.17 | .03 | **.44** | **.40** | **.40** | **.54** | 1.00 |

ESEM = exploratory structural equation modeling; i01-i36 = items; F1-F9 = factors; Factor loadings ≥ .40 in absolute value are bolded and highlighted in grey. Within the correlation matrix, those correlations that show a decrease in magnitude (Δ ≥ .10) have been highlighted in boldface, comparing in all cases with the interfactor correlations derived from the CFA. In the case of the correlation matrix corresponding to the Set-ESEM, those correlations that account for an increase in magnitude (Δ ≥ .10) have also been highlighted in bold italics, comparing in all cases with the interfactor correlations derived from the CFA. Cross-loadings fixed to zero appear in italics. p < .05 for all factor loadings and factor correlations, except those underlined.

theoretically incongruent correlation. Here, a positive correlation of low magnitude is evident between the factors Rumination and Acceptance. In the Full-ESEM model with oblique rotation, no theoretically incongruent correlation between the latent factors is evident. Likewise, in the Full-ESEM model with oblique rotation for most of the interfactor correlations described a larger decrease in the magnitude of these correlations is observed (Table 3).

Regarding the standardized regression weights estimated in each model, it can be generally stated that there is a great variation in the magnitude of these parameters depending on the model specified (CFA model vs. ESEM models; see Tables 2 and 3). For example, in the CFA model, item 25 presents a standardized regression weight of −.21 in its respective factor, while in the Set-ESEM model, this item presents a standardized regression weight of −.75 with the reinterpretation factor. In the case of the Full-ESEM models, item 25 presents standardized regression weights between .28 and .27 with its factor, also showing cross-loadings of similar magnitude with other factors. Another example is item 26, which in the Set-ESEM and Full-ESEM model with oblique rotation presents a standardized regression weight of .22 and .24 with its respective factor (self-blame), being higher with the rumination factor (.45 and .50). This same item does not present a standardized regression weight higher than .40 with any factor when a Full-ESEM model with target rotation is specified.

Regarding the evidence of cross-loadings of indicators with different latent factors, it is possible to verify in the data of Tables 2 and 3 that when less restrictive models are specified, some items present factor loadings higher and above .40 with latent factors different from what was suggested by the original nine-factor model. Furthermore, in some situations, these standardized regression weights are greater than .40 with more than one latent factor. In conclusion: 1- as the models become less restrictive, the anomalous results (i.e., theoretically incongruent interfactor correlations) disappear; 2- in the less restrictive models' the interfactor correlations decrease in magnitude, and this decrease is directly proportional to the degree of restriction of the specified model; 3- in the less restrictive models a greater degree of cross-loadings can be observed. Finally, contrary to the hypothesis proposed in the present work, the Set-ESEM model did not present a more adequate solution or fit compared to less restrictive models, which also suggests that the cross-loadings of the indicators with different latent factors are not circumscribed to the groups of CERS described.

**Table 3. Standardized regression weights for Full-ESEM models with Target and Oblique rotation.**

| Ítem/Factor | Full-ESEM Target | | | | | | | | | Full-ESEM Oblique | | | | | | | | |
|---|---|---|---|---|---|---|---|---|---|---|---|---|---|---|---|---|---|---|
| | F1 | F2 | F3 | F4 | F5 | F6 | F7 | F8 | F9 | F1 | F2 | F3 | F4 | F5 | F6 | F7 | F8 | F9 |
| **S-B** | | | | | | | | | | | | | | | | | | |
| i01 | **.73** | −.01 | .02 | −.02 | .00 | −.00 | .00 | −.08 | −.03 | **.66** | .11 | .02 | −.01 | −.00 | −.01 | .00 | −.08 | −.06 |
| i17 | **1.04** | −.16 | −.03 | .00 | −.01 | .03 | −.01 | .03 | .02 | **.95** | −.01 | −.01 | .02 | −.01 | .01 | −.00 | .00 | .02 |
| i26 | .30 | .38 | .06 | −.03 | .06 | −.04 | .01 | .02 | .05 | .22 | **.50** | .01 | −.05 | .04 | −.02 | .01 | .03 | .05 |
| i33 | **.57** | .12 | .08 | −.05 | .05 | −.00 | .01 | .08 | −.04 | **.50** | .25 | .05 | −.05 | .05 | .00 | .01 | .08 | −.04 |
| **Rum.** | | | | | | | | | | | | | | | | | | |
| i03 | .07 | **.48** | .04 | .06 | .01 | −.07 | −.00 | −.06 | .15 | .01 | **.58** | −.01 | .03 | −.00 | −.06 | −.01 | −.03 | .15 |
| i15 | .14 | **.45** | .17 | .02 | −.01 | −.10 | −.00 | −.06 | .08 | .08 | **.58** | .10 | .00 | −.03 | −.08 | −.00 | −.04 | .07 |
| i18 | .14 | **.49** | .00 | .04 | .03 | −.00 | .03 | −.01 | .17 | .07 | **.59** | −.05 | .01 | .01 | .00 | .02 | .00 | .17 |
| i27 | .04 | **.70** | .15 | .01 | .02 | −.01 | −.03 | −.02 | −.02 | −.04 | **.83** | .04 | −.02 | .00 | .02 | −.03 | .02 | −.04 |
| **Catas.** | | | | | | | | | | | | | | | | | | |
| i08 | .02 | −.17 | **.64** | .02 | −.08 | −.01 | .14 | .03 | .06 | .04 | −.03 | **.60** | .02 | −.07 | −.01 | .18 | −.00 | .09 |
| i10 | .13 | .18 | **.49** | .08 | −.00 | −.10 | −.07 | −.10 | −.01 | .09 | .36 | **.42** | .06 | −.01 | −.08 | −.04 | −.08 | −.03 |
| i22 | −.02 | −.06 | **.82** | .03 | −.01 | .09 | −.10 | .05 | −.04 | −.01 | .11 | **.74** | .02 | −.02 | .10 | −.06 | .04 | −.02 |
| i35 | .05 | .31 | **.49** | .05 | .04 | −.07 | −.01 | −.10 | −.07 | .01 | **.49** | .41 | .02 | .02 | −.04 | .00 | −.06 | −.09 |
| **O-B** | | | | | | | | | | | | | | | | | | |
| i09 | −.02 | −.06 | −.03 | **.82** | .02 | −.02 | −.02 | −.06 | .07 | −.00 | −.04 | −.00 | **.81** | .01 | −.03 | −.00 | −.06 | .05 |
| i21 | −.02 | .13 | .06 | **.52** | −.06 | −.03 | .18 | .08 | −.01 | −.03 | .18 | .03 | **.51** | −.06 | −.02 | .20 | .06 | −.01 |
| i29 | .01 | −.04 | −.02 | **.93** | .04 | .02 | −.08 | .02 | −.01 | .02 | −.01 | −.00 | **.91** | .03 | .02 | −.05 | .01 | −.03 |
| i36 | −.04 | −.00 | .06 | **.75** | .00 | .01 | .00 | .00 | −.00 | −.03 | .03 | .07 | **.74** | −.00 | .01 | .03 | −.00 | −.01 |
| **Acept.** | | | | | | | | | | | | | | | | | | |
| i02 | .08 | −.09 | −.09 | .00 | **.66** | −.03 | −.00 | −.09 | .06 | .09 | −.05 | −.05 | .01 | **.64** | −.03 | −.01 | −.02 | .06 |
| i16 | −.01 | −.04 | −.07 | −.00 | **.75** | −.00 | .04 | .02 | .06 | .00 | −.01 | −.03 | .00 | **.72** | −.00 | .03 | .08 | .08 |
| i32 | −.01 | .09 | .00 | .00 | **.57** | .02 | .03 | .19 | −.05 | −.01 | .13 | .00 | −.00 | **.54** | .04 | .03 | .22 | −.01 |
| i25 | .00 | .11 | .22 | .06 | .28 | .11 | .01 | −.12 | −.32 | −.00 | .17 | .19 | .04 | .27 | .13 | .03 | −.04 | −.37 |
| **Ref.** | | | | | | | | | | | | | | | | | | |
| i04 | −.01 | −.09 | −.02 | −.00 | .01 | **.67** | .01 | −.02 | .13 | −.00 | −.13 | −.00 | −.00 | .01 | **.64** | .00 | −.01 | .15 |
| i14 | .00 | .00 | .04 | −.00 | .01 | **.88** | .01 | −.10 | .05 | −.00 | −.01 | .04 | −.00 | .01 | **.85** | .00 | −.06 | .03 |
| i24 | .02 | .00 | −.00 | .00 | −.00 | **.86** | .02 | .04 | −.06 | .02 | −.03 | −.00 | −.00 | −.00 | **.84** | .02 | .06 | −.05 |
| i28 | −.00 | .07 | −.02 | .01 | −.00 | **.78** | .01 | .09 | .01 | −.01 | .03 | −.03 | .00 | −.00 | **.76** | .00 | .10 | .03 |
| **Pers.** | | | | | | | | | | | | | | | | | | |
| i07 | .02 | −.14 | .17 | −.03 | .02 | −.02 | **.50** | .09 | .04 | .03 | −.10 | .17 | −.02 | .03 | −.02 | **.50** | .06 | .07 |
| i11 | −.03 | .00 | .00 | .00 | .02 | −.00 | **.80** | −.13 | .00 | −.03 | .01 | .00 | −.00 | .03 | −.00 | **.80** | −.12 | −.01 |
| i20 | .03 | −.00 | −.13 | .05 | −.06 | .05 | **.66** | .06 | .06 | .02 | −.03 | −.11 | .05 | −.05 | .04 | **.66** | .03 | .08 |
| i34 | −.00 | .06 | −.02 | −.01 | .08 | .03 | **.72** | .05 | −.08 | −.01 | .06 | −.03 | −.02 | .09 | .04 | **.72** | .05 | −.07 |
| **Reint.** | | | | | | | | | | | | | | | | | | |
| i06 | −.02 | −.11 | .10 | −.07 | .10 | −.01 | −.00 | **.55** | .26 | −.01 | −.08 | .09 | −.05 | .09 | −.01 | −.01 | **.45** | **.40** |
| i12 | −.12 | −.04 | .07 | −.03 | .14 | .00 | .08 | **.47** | .25 | −.10 | −.02 | .06 | −.02 | .13 | .01 | .08 | .38 | .37 |
| i23 | .06 | .03 | −.09 | .05 | −.04 | −.03 | .04 | **.96** | −.14 | .04 | .00 | −.13 | .05 | −.04 | .00 | .04 | **.82** | .01 |
| i31 | −.00 | .01 | −.06 | −.00 | .02 | .14 | .01 | **.75** | .00 | −.01 | −.01 | −.09 | .00 | .02 | .16 | .00 | **.64** | .13 |
| **F-P** | | | | | | | | | | | | | | | | | | |
| i05 | −.03 | −.07 | .00 | −.01 | .10 | .07 | .02 | .05 | **.59** | −.02 | −.04 | .02 | −.00 | .09 | .03 | .01 | .00 | **.66** |
| i13 | −.10 | .04 | .00 | −.03 | .08 | .09 | .03 | .16 | **.59** | −.09 | .07 | .00 | −.02 | .07 | .06 | .01 | .10 | **.68** |
| i19 | .08 | .21 | −.05 | .01 | −.05 | .05 | .10 | .01 | **.61** | .05 | .28 | −.06 | .01 | −.06 | .02 | .07 | −.03 | **.67** |
| i30 | −.00 | .09 | −.05 | .06 | .03 | .10 | −.00 | .16 | **.56** | −.01 | .13 | −.05 | .07 | .01 | .07 | −.02 | .10 | **.64** |

*(Continued)*

**Table 3.** (Continued)

| Ítem/Factor | Full-ESEM Target | | | | | | | | | Full-ESEM Oblique | | | | | | | | |
|---|---|---|---|---|---|---|---|---|---|---|---|---|---|---|---|---|---|---|
| | F1 | F2 | F3 | F4 | F5 | F6 | F7 | F8 | F9 | F1 | F2 | F3 | F4 | F5 | F6 | F7 | F8 | F9 |
| **Interfactor Correlation** | F1 | F2 | F3 | F4 | F5 | F6 | F7 | F8 | F9 | F1 | F2 | F3 | F4 | F5 | F6 | F7 | F8 | F9 |
| F1 | 1.00 | | | | | | | | | 1.00 | | | | | | | | |
| F2 | **.48** | 1.00 | | | | | | | | **.45** | 1.00 | | | | | | | |
| F3 | **.40** | **.44** | 1.00 | | | | | | | **.25** | **.37** | 1.00 | | | | | | |
| F4 | .01 | .21 | .38 | 1.00 | | | | | | −.08 | .23 | .34 | 1.00 | | | | | |
| F5 | .01 | .09 | **−.01** | −.06 | 1.00 | | | | | −.01 | .02 | **−.09** | −.08 | 1.00 | | | | |
| F6 | −.33 | −.23 | **−.22** | −.02 | **.34** | 1.00 | | | | −.26 | −.26 | **−.18** | .01 | **.32** | 1.00 | | | |
| F7 | −.06 | −.03 | −.18 | .02 | **.43** | .45 | 1.00 | | | −.05 | −.00 | −.18 | .01 | **.40** | .45 | 1.00 | | |
| F8 | −.27 | −.15 | **−.10** | −.14 | **.48** | .61 | .57 | 1.00 | | −.19 | −.23 | **−.29** | −.11 | **.41** | **.55** | .56 | 1.00 | |
| F9 | −.08 | .07 | −.34 | −.05 | **.39** | **.32** | **.32** | **.60** | 1.00 | −.13 | −.09 | −.32 | −.08 | **.41** | **.42** | **.38** | **.59** | 1.00 |

ESEM = exploratory structural equation modeling; i01-i36 = items; F1-F9 = factors; Factor loadings ≥ .40 in absolute value are bolded and highlighted in grey. Within the correlation matrix, correlations showing a decrease in magnitude (Δ ≥ .10) have been highlighted in bold, comparing in all cases with the interfactor correlations derived from the CFA. Cross-loadings fixed to zero appear in italics. $p < .05$ for all factor loadings and factor correlations, except those underlined.

## Discussion

The present research aimed to apply ESEM models to study the internal structure of CERQ-derived measures. Two hypotheses were proposed: 1- in comparison with the CFA Model, in the ESEM models, the parameters will be better estimated, expressed especially in the standardized regression weights and the interfactor correlations; and 2- within the ESEM models, in the Set-ESEM model the parameters will be correctly estimated evidencing a more restricted solution. Results demonstrated partial evidence in favor of hypothesis number 1, while the results did not support hypothesis number 2.

Regarding the first hypothesis, in the CFA model, there were high-magnitude interfactor correlations; in the less restrictive models (i.e., ESEM), these correlations decreased in magnitude; the decrease in interfactor correlations is accompanied by evidence of cross-loadings in several indicators. Even so, both Set-ESEM models with oblique rotation and Full-ESEM with target rotation showed anomalous results, and the Set-ESEM model with target rotation could not be estimated due to model identification problems. In this sense, there was no evidence supporting the second hypothesis. The latter suggests that the cross-loadings of the various indicators do not only occur between latent factors of the same group of CERS but involve the whole set of strategies considered. From all this, it is concluded that, although for the CERQ-derived measures, some parameters are better estimated in the ESEM models, the model that shows the best fit is the least restrictive, the Full-ESEM model with oblique rotation. Unlike the other ESEM models, no anomalous results are found in the latter model, the interfactor correlations decrease in greater magnitude, and there is evidence of cross-loadings of indicators that show some overlap in their content. All this suggests that, in the case of CERQ measurements, we would not be in the presence of indicators that reflect simple factorial structures [22], at least in most cases, as is explained below.

Given the better fit of the Full-ESEM model with oblique rotation, the discussion will focus on the parameters estimated from that model. First, it is important to discuss the interfactor correlations evidenced in the CFA model and those resulting from the Full-ESEM model with oblique rotation. Visualizing only the CFA model, there seems to be a high correlation between the variables self-blame and catastrophizing. However, this correlation becomes low in the Full-ESEM model. Another similar example is the correlation between catastrophizing and acceptance. While in the CFA model, a negative correlation of low to moderate magnitude (−.27) is observed, in the Full-ESEM model with oblique rotation this negative correlation is very low (−.09). Similarly, the high magnitude correlations observed in the CFA model between the factors

rumination and catastrophizing (.75), catastrophizing and positive refocusing (−.40), catastrophizing and positive reinterpretation (−.51) can be mentioned and considered; and the differences that are appreciated, in decreasing magnitude, with the Full-ESEM model with oblique rotation (r rumination and catastrophizing = .37; r catastrophizing and positive refocusing = −.18; r catastrophizing and positive reinterpretation = −.29). A similar situation is present in the interfactor correlations between the latent factors acceptance, positive refocusing, putting into perspective, positive reinterpretation, and refocusing on plans. The only exception is the indicators referring to the factor blaming others, where there were no changes in the estimated interfactor correlations. It is important to note that, to date, analyses of the internal structure of the CERQ have relied exclusively on confirmatory factor analysis (CFA) [8,12,14,16–18]. In this study, we show that, in the case of CERQ-derived measurements, the application of CFA models often results in biased parameter estimates, particularly through the overestimation of correlations among different cognitive emotion regulation strategies (CERS).

Secondly, data of interest are also verified when considering the estimated standardized regression weights and the comparison between the CFA model and the Full-ESEM model with oblique rotation. In the latter case, the presence of cross-loadings is verified as well as standardized regression weights greater than or equal to .40 with latent factors that do not correspond to the original 9-factor proposal. Specifically, this performance is corroborated in the following cases: item 26 (standardized regression weight of .50 with the rumination factor, and .22 with its self-blame factor), item 35 (standardized regression weight of .49 with the rumination factor, and .41 with its catastrophizing factor), and item 6 (standardized regression weight of .45 with its positive reinterpretation factor, and .40 with its refocusing on plans factor). In addition, if we consider a less conservative cut-off point, such as standardized regression weights greater than or equal to .35, we add more evidence of cross-loadings and standardized regression weights with latent factors that do not correspond to the original 9-factor proposal, such as: item 12 (standardized regression weight of .38 with its reinterpretation factor), item 12 (standardized regression weight of .38 with its positive reinterpretation factor, and .37 with its refocusing on plans factor), item 25 (standardized regression weight of .27 with its acceptance factor, and −.37 with its refocusing on plans factor), and item 10 (standardized regression weight of .42 with its catastrophizing factor and .36 with its rumination factor). Only three of the nine latent factors originally proposed do not present this type of behavior: blaming others, positive refocusing, and putting into perspective. Only in these cases would there be evidence favoring a simple factor structure. The opposite situation evidenced for the rest of the latent factors and their indicators would be explained in terms of content validity failures, probably due to: a) indicators whose content does not correspond to the proposed construct definition (e.g., item 25); b) content overlap between indicators (e.g., item 6, item 26, item 35). These results are consistent with the conclusions we previously reached in the introduction regarding the existing evidence on the psychometric properties of the CERQ. A review of these prior investigations shows that the original nine-factor model does not demonstrate adequate fit to the data. In this sense, different studies have found that the original nine-factor model only achieves an acceptable fit after several post hoc modifications, such as correlating residual errors [8–12], removing several items [13], or specifying the cross-loading of certain items on theoretically distinct factors [9,11]. Overlap between some latent factors, particularly Rumination and Catastrophizing, has also been observed [4,11,13]. Likewise, poor fit indicators have been reported, such as low factor loadings (<.30), or non-significant and negative factor loadings for some items within its factors [9,13,14].

All the above gives rise to questions that have important practical implications. In this sense, it is worth asking how the estimation of parameters with a very restrictive model (i.e., CFA) in cases such as the one we present from CERQ can lead not only to biasing the resulting parameters in a measurement model but also to obtaining biased estimates in regression or structural models where other variables are considered for predictive purposes. Along these lines, some studies have highlighted the importance of adequate factor modelling to estimate factor scores and the use of these scores for predictive purposes [38]. Also, the ESEM is recommended for multiple regression with latent variables when nonignorable cross-factor loadings exist [39]. All of this becomes more relevant when considering that much of the applied literature has based its correlation/prediction studies on the premise that a simply structured CFA model (i.e., original 9-factor model) is the best-fit model to account for CERQ scores [40].

Regarding limitations, this work has not analyzed the criterion validity with variables derived from other instruments. It would be important for future studies to determine whether the overestimation of the parameters evidenced in the case of the CERQ is also observed when a structural model is specified, including other measurement models derived from different instruments. For these purposes, the combined use of other types of derived measurements is relevant, for example, behavioral measures or performance-based tests (e.g., an ability test of emotional intelligence, such as the Mayer-Salovey-Caruso Emotional Intelligence Test, [41]). On the other hand, although we have worked with a large sample, in the present work, we have not applied analyses that would allow us to demonstrate the stability of the results obtained. Future work could advance on this point by incorporating relevant analyses (e.g., bootstrapping methods; cross-validity analysis between independent samples) or by replicating the study in populations different from the present study. Additionally, the replicability of the present results could be examined through the implementation of longitudinal and repeated measures designs (e.g., ecological momentary assessment; see [42]).

## Conclusion

Based on the results of the present study, it can be concluded that the simple 9-factor structure traditionally proposed for the CERQ [7,10] does not exhibit adequate model fit. This finding suggests the need for caution among applied psychologists and clinical researchers who interpret scale scores as if they reflect distinct, unidimensional factors. The common practice of generating summed scores across items associated with each of the originally proposed factors may therefore be questionable. The evidence presented here suggests that in the case of the CERQ, there would be flaws in the content validity of some of its indicators and that factor modelling of the scores derived from this instrument using CFA results in biased parameter estimates that greatly overestimate the interfactor correlations. Exceptions to this general conclusion are the indicators reflecting the latent variables of blaming others, positive refocusing, and putting into perspective, for which evidence of simple factor structure has been obtained here. Researchers interested in the use of CERQ may choose to use a strategy similar to that applied by Heinrich et al. [43]. There, it is suggested, first, to start from the content validity of the indicators that would be reflective of the latent factors of interest. In the case of CERQ, it is suggested to consider only those indicators per latent factor that have not presented evidence of cross-loadings or loadings attainable on factors other than those originally proposed (according to the evidence presented here, in the case of self-blaming, for example, only items 1, 17 and 33 would be considered). Then, Henrich et al. [43] exemplify the use of bifactor (S-1) models to test predictive hypotheses. They suggested that this type of measurement model is relevant for multidimensional scales, as is the case of the CERQ. Another alternative possibility is the in-depth review of the content validity of the indicators included in the CERQ, particularly those whose scores show complex behavior. All in all, it is hoped that the data offered by the present work will make it possible to achieve greater coherence between the psychometric evidence and the applied use of the CERQ measures. A more nuanced understanding of the questionnaire's internal structure may ultimately enhance the reliability and interpretability of CERQ scores in both clinical and applied settings, thereby improving the quality of psychological assessment and intervention outcomes.

## Author contributions

**Conceptualization:** Pablo Ezequiel Flores-Kanter, Leonardo Medrano.

**Data curation:** Pablo Ezequiel Flores-Kanter.

**Formal analysis:** Pablo Ezequiel Flores-Kanter.

**Investigation:** Pablo Ezequiel Flores-Kanter.

**Methodology:** Pablo Ezequiel Flores-Kanter.

**Supervision:** Luciana Moretti, Zoilo Emilio García-Batista, Leonardo Medrano.

**Validation:** Leonardo Medrano.

**Writing – original draft:** Pablo Ezequiel Flores-Kanter.

**Writing – review & editing:** Pablo Ezequiel Flores-Kanter, Luciana Moretti, Zoilo Emilio García-Batista, Leonardo Medrano.

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
