## [Decision Letter · Decision Letter 0]

Dear Dr. Flores-Kanter,

Thank you for submitting your manuscript to PLOS ONE. After careful consideration, we feel that it has merit but does not fully meet PLOS ONE’s publication criteria as it currently stands. Therefore, we invite you to submit a revised version of the manuscript that addresses the points raised during the review process.

We look forward to receiving your revised manuscript.

Kind regards,

Diogo Manuel Teixeira Monteiro, PhD

Academic Editor

PLOS ONE

Reviewers' comments:

Reviewer's Responses to Questions

**Comments to the Author**

1. Is the manuscript technically sound, and do the data support the conclusions?

Reviewer #1: Yes

Reviewer #2: Yes

2. Has the statistical analysis been performed appropriately and rigorously?

Reviewer #1: Yes

Reviewer #2: Yes

3. Have the authors made all data underlying the findings in their manuscript fully available?

Reviewer #1: Yes

Reviewer #2: Yes

4. Is the manuscript presented in an intelligible fashion and written in standard English?

Reviewer #1: Yes

Reviewer #2: Yes

Reviewer #1: In this study, Flores-Kanter et al. proposed to estimate innovative measurement models for the Cognitive Emotion Regulation Questionnaire, ie., using the exploratory structural equation models approaches. The authors present a correct and transparent methodology, however, some minor issues need clarification. Authors are invited to address the points outlined below:

1. The abstract should be revised to clearly state the objective of the study. The statement “worked with a large sample of Argentines” is vague; indicate the sample size and mean age of participants. In the results section, it would be better to mention how much better the ESEM models performed (please include some numbers to provide precise information).

2. The Methods section should clearly state any inclusion or exclusion criteria applied during data collection, i.e., incomplete responses excluded or duplicate responses.

3. If available, include more demographic details (i.e., sex, education, socioeconomic status) to assess sample representativeness.

4. Please report basic psychometric properties (i.e., internal consistency) of the Argentine adaptation of the CERQ (Medrano et al., 2013).

5. L207 - the authors stated that the Set-ESEM model with Target rotation "could not be identified," but do not explain why.

6. The paragraph of limitations should be completed by discussing possible cultural specificity of the findings and acknowledging potential biases associated with self-reported data.

7. I suggest grouping future research directions into a coherent final paragraph.

8. References are not in accordance with the journal's guidelines. PLOS uses the reference style outlined by the International Committee of Medical Journal Editors (ICMJE), also referred to as the “Vancouver” style.

Reviewer #2: Thank you for the opportunity to review the manuscript entitled “Cognitive Emotion Regulation Questionnaire: Evidence of Internal Structure through Confirmatory Factor Modeling and Exploratory Structural Equation Modeling”.

The study addresses a relevant topic and presents a rigorous methodological analysis. However, some improvements are needed to strengthen the clarity and presentation of the results. I present my considerations and suggestions below:

1. The summary ends abruptly. I recommend that the authors add a final sentence highlighting the practical contributions of the findings.

2. The introduction is excessively dense and difficult to read. I suggest that the authors simplify the wording and make the exposition more objective, making it easier for the reader to understand.

3. It is important to expand on the justification for carrying out this study, highlighting the gaps in the literature that the research seeks to fill.

4. I would ask the authors to clarify how they theoretically justify the division into two large groups of strategies (CERS group 1 vs. group 2) in the Set-ESEM model.

5. I recommend that the authors discuss more explicitly how their findings relate to previous cross-cultural studies that have used the CERQ.

6. I ask for clarification as to whether there was a prior calculation of the sample size required for the study.

7. The instruments used could be described in more detail and depth, including information on their psychometric properties in similar populations.

8. I suggest that the inclusion and exclusion criteria used to select the participants be explicitly presented.

9. I have not found information on the approval of the study by a human research ethics committee. I recommend that this information be included, as required by international research standards.

10. Although the manuscript addresses some limitations, I suggest that the authors broaden this discussion, indicating practical ways in which future research can overcome them.

11. Tables 2 and 3 are excessively long and difficult to interpret. I recommend presenting summaries or breaking down the information, focusing on the most relevant items (for example, those with significant cross-loadings).

12. The authors could further explore the practical implications of the results for the use of CERQ in clinical and research contexts.

13. I recommend including a “Conclusion” section, summarizing the main findings and their practical and theoretical implications.

**Do you want your identity to be public for this peer review?** For information about this choice, including consent withdrawal, please see our Privacy Policy

Reviewer #1: No

Reviewer #2: **Yes: ** Miguel Jacinto

---

## [Author Response · Author response to Decision Letter 1]

14 May 2025

We would like to express our gratitude the editor for providing us with the opportunity to respond to the reviewer's comments. We also extend our appreciation to both the editor and the reviewer for their time and effort in reviewing our manuscript. Below, we provide point-by-point responses to the comments, and the modifications made have been tracked in the manuscript.

Abstract

Reviewer #1: The statement “worked with a large sample of Argentines” is vague; indicate the sample size and mean age of participants.

Response: Thank you. We changed the sentence accordingly (line 7-10): “For this purpose, we worked with a large sample of Argentines of 6881 Argentine adults aged between 18 and 81 (M = 27.14, SD = 9.86; 69.6% female; 86.4% not being in psychological or psychiatric treatment) and compared the fit of the models.”

Reviewer #1: In the results section, it would be better to mention how much better the ESEM models performed (please include some numbers to provide precise information).

Response: Thank you. The suggested information has been added (line 10-12): “The results favored the exploratory structural equation models (χ2 = 4092.89, df = 342, CFI = .982, SRMR = .013, RMSEA = .04), indicating that we are not observing indicators that reflect simple factorial structures.”

Reviewer #2: The summary ends abruptly. I recommend that the authors add a final sentence highlighting the practical contributions of the findings.

Response: Thank you. The suggested information has been added (line 12-19): “Based on the results of the present study, it can be concluded that the simple 9-factor structure traditionally proposed for the CERQ does not exhibit adequate model fit. This finding suggests the need for caution among applied psychologists and clinical researchers who interpret scale scores as if they reflect distinct, unidimensional factors. The common practice of generating summed scores across items associated with each of the originally proposed factors may therefore be questionable. A more nuanced understanding of the questionnaire's internal structure may ultimately enhance the reliability and interpretability of CERQ scores in both clinical and applied settings, thereby improving the quality of psychological assessment and intervention outcomes.”

Introduction

Reviewer #2: The introduction is excessively dense and difficult to read. I suggest that the authors simplify the wording and make the exposition more objective, making it easier for the reader to understand.

Response: Thank you. We simplified the wording and tried to make the exposition more direct (see lines 22–69).

Reviewer #2: It is important to expand on the justification for carrying out this study, highlighting the gaps in the literature that the research seeks to fill.

Response: Thank you. The suggested information has been added (see lines 22–69).

Methods

Reviewer #2: I ask for clarification as to whether there was a prior calculation of the sample size required for the study.

Response: Thank you. The suggested information has been added (line 73-75): “Although a prior calculation of the required sample size was not conducted, a sufficiently large sample was anticipated to ensure the stability and accuracy of the estimates derived from the applied models.”

Reviewer #1: If available, include more demographic details (i.e., sex, education, socioeconomic status) to assess sample representativeness.

Response: Thank you. The suggested information has been added (line 75-77): “The sample consisted of 6881 Argentine adults aged between 18 and 81 (M = 27.14, SD = 9.86). Of the total sample, 69.6% (n = 4,792) identified as female, while 86.4% (n = 5,948) reported not being in psychological or psychiatric treatment.”

Reviewer #2: I have not found information on the approval of the study by a human research ethics committee. I recommend that this information be included, as required by international research standards.

Response: Thank you. This information can be found in lines 82-83: “The ethics committee of the Research Secretariat of the Universidad Siglo 21 previously approved the research protocol following the ethical guidelines of the APA.”

Reviewer #1: The Methods section should clearly state any inclusion or exclusion criteria applied during data collection, i.e., incomplete responses excluded or duplicate responses. Reviewer #2: I suggest that the inclusion and exclusion criteria used to select the participants be explicitly presented.

Response: Thank you. The suggested information has been added (line 87-88): “It is important to note that, given the data collection method, there were no missing data, and the only inclusion criteria were agreeing to participate in the study and being 18 years of age or older.”

Reviewer #1: Please report basic psychometric properties (i.e., internal consistency) of the Argentine adaptation of the CERQ (Medrano et al., 2013). Reviewer #2: The instruments used could be described in more detail and depth, including information on their psychometric properties in similar populations

Response: Thank you. The suggested information has been added (line 97-101): “The Argentine version of the scale was used (34). This version was adapted and validated in a sample of university students (N = 359, M age = 24.6, female = 50.1). The model of 9 correlated factors showed acceptable fit indicators (χ2 = 875.50, df = 538, CFI = .91, GFI = .90, RMSEA = .04), and the internal consistency indicators measured by Cronbach’s Alpha varied between .59 and .83.”

Reviewer #2: I would ask the authors to clarify how they theoretically justify the division into two large groups of strategies (CERS group 1 vs. group 2) in the Set-ESEM model.

Response: Thank you. We added this clarification in lines 103-126.

Results

Reviewer #2: Tables 2 and 3 are excessively long and difficult to interpret. I recommend presenting summaries or breaking down the information, focusing on the most relevant items (for example, those with significant cross-loadings).

Response: Thank you. We have not modified the information presented in Tables 2 and 3, as all content is pertinent and relevant for a proper understanding of the results. Moreover, a comprehensive presentation is essential to ensure the reproducibility of the findings.

Discussion

Reviewer #2: I recommend that the authors discuss more explicitly how their findings relate to previous cross-cultural studies that have used the CERQ.

Response: Thank you. We have added a more explicit discussion of how our findings relate to previous cross-cultural studies that have employed the CERQ. Please see lines 266-271 and 292-300.

Reviewer #1: the authors stated that the Set-ESEM model with Target rotation "could not be identified," but do not explain why.

Response: Thank you. This explanation can be found in lines 239-250: “Even so, both Set-ESEM models with oblique rotation and Full-ESEM with target rotation showed anomalous results, and the Set-ESEM model with target rotation could not be estimated due to model identification problems. In this sense, there was no evidence supporting the second hypothesis. The latter suggests that the cross-loadings of the various indicators do not only occur between latent factors of the same group of CERS but involve the whole set of strategies considered. From all this, it is concluded that, although for the CERQ-derived measures, some parameters are better estimated in the ESEM models, the model that shows the best fit is the least restrictive, the Full-ESEM model with oblique rotation. Unlike the other ESEM models, no anomalous results are found in the latter model, the interfactor correlations decrease in greater magnitude, and there is evidence of cross-loadings of indicators that show some overlap in their content. All this suggests that, in the case of CERQ measurements, we would not be in the presence of indicators that reflect simple factorial structures (Morin et al., 2015), at least in most cases, as is explained below.”

Reviewer #1: The paragraph of limitations should be completed by discussing possible cultural specificity of the findings and acknowledging potential biases associated with self-reported data. Reviewer #1: I suggest grouping future research directions into a coherent final paragraph. Reviewer #2: Although the manuscript addresses some limitations, I suggest that the authors broaden this discussion, indicating practical ways in which future research can overcome them.

Response: Thank you. The suggested information has been added (line 312-317 and line 321-323): “It would be important for future studies to determine whether the overestimation of the parameters evidenced in the case of the CERQ is also observed when a structural model is specified, including other measurement models derived from different instruments. For these purposes, the combined use of other types of derived measurements is relevant, for example, behavioral measures or performance-based tests (e.g., an ability test of emotional intelligence, such as the Mayer-Salovey-Caruso Emotional Intelligence Test, 43).” And “Additionally, the replicability of the present results could be examined through the implementation of longitudinal and repeated measures designs (e.g., ecological momentary assessment; see 44).”

Reviewer #2: The authors could further explore the practical implications of the results for the use of CERQ in clinical and research contexts. I recommend including a “Conclusion” section, summarizing the main findings and their practical and theoretical implications.

Response: Thank you. We added a Conclusion section in which we summarize the main findings and explore the practical implications of the results for the use of the CERQ in clinical and research contexts.

---

## [Decision Letter · Decision Letter 1]

Dear Dr. Flores-Kanter,

Thank you for submitting your manuscript to PLOS ONE. After careful consideration, we feel that it has merit but does not fully meet PLOS ONE’s publication criteria as it currently stands. Therefore, we invite you to submit a revised version of the manuscript that addresses the points raised during the review process.

Please submit your revised manuscript by Jul 04 2025 11:59PM. If you will need more time than this to complete your revisions, please reply to this message or contact the journal office at plosone@plos.org . A rebuttal letter that responds to each point raised by the academic editor and reviewer(s). You should upload this letter as a separate file labeled 'Response to Reviewers'.A marked-up copy of your manuscript that highlights changes made to the original version. You should upload this as a separate file labeled 'Revised Manuscript with Track Changes'.An unmarked version of your revised paper without tracked changes. You should upload this as a separate file labeled 'Manuscript'.

We look forward to receiving your revised manuscript.

Kind regards,

Diogo Manuel Teixeira Monteiro, Ph.D.

Academic Editor

PLOS ONE

Journal Requirements:

Reviewers' comments:

Reviewer's Responses to Questions

**Comments to the Author**

Reviewer #1: All comments have been addressed

Reviewer #2: All comments have been addressed

2. Is the manuscript technically sound, and do the data support the conclusions?

Reviewer #1: Yes

Reviewer #2: Yes

3. Has the statistical analysis been performed appropriately and rigorously?

Reviewer #1: Yes

Reviewer #2: Yes

4. Have the authors made all data underlying the findings in their manuscript fully available?

Reviewer #1: Yes

Reviewer #2: Yes

5. Is the manuscript presented in an intelligible fashion and written in standard English?

Reviewer #1: Yes

Reviewer #2: Yes

Reviewer #1: (No Response)

Reviewer #2: Since the sample size has not been calculated, it is necessary to understand how the authors ensure that the sample is sufficiently large.

The eligibility criteria for the participants are still not presented.

**Do you want your identity to be public for this peer review?** For information about this choice, including consent withdrawal, please see our Privacy Policy

Reviewer #1: No

Reviewer #2: **Yes: ** Miguel Jacinto

---

## [Author Response · Author response to Decision Letter 2]

27 May 2025

We would like to express our gratitude the editor for providing us with the opportunity to respond to the reviewer's comments. We also extend our sincere appreciation to both the editor and the reviewer for their time and effort in re-evaluating our manuscript. Below, we provide point-by-point responses to the comments, and the modifications made have been tracked in the manuscript.

Reviewer #2: Since the sample size has not been calculated, it is necessary to understand how the authors ensure that the sample is sufficiently large.

Response: Thank you. The suggested information has been added (see lines 86–92): “Although a prior calculation of the required sample size was not conducted, a sufficiently large sample was anticipated to ensure both the stability (i.e., that the estimation algorithm would converge on an admissible solution) and the accuracy of the parameter estimates (i.e., that the estimated parameter values would not substantially differ from the true population values) derived from the tested models (26,27). The simulation studies reported by Wolf et al. (26) suggest a minimum required sample size of 500 participants, considering the complexity of the models evaluated here (e.g., number of parameters to be estimated, number of items per factor).”

Reviewer #2: The eligibility criteria for the participants are still not presented.

Response: Thank you. This information can be found in lines 105-106: “It is important to note that, given the data collection method, there were no missing data, and the only inclusion criteria were agreeing to participate in the study and being 18 years of age or older.”

Journal Requirements:

Please review your reference list to ensure that it is complete and correct.

Response: Thank you. We have carefully reviewed the reference list to ensure it is complete and correctly formatted in accordance with the journal's requirements.

Thanks again, and we look forward to hearing from you.

---

## [Editor Report · Decision Letter 2]

Cognitive Emotion Regulation Questionnaire: Evidence of Internal Structure through Confirmatory Factor Modeling and Exploratory Structural Equation Modeling

PONE-D-25-17586R2

Dear Dr. Flores-Kanter,

We’re pleased to inform you that your manuscript has been judged scientifically suitable for publication and will be formally accepted for publication once it meets all outstanding technical requirements.

Kind regards,

Diogo Manuel Teixeira Monteiro, Ph.D.

Academic Editor

PLOS ONE
---

## [Editor Report · Acceptance letter]

PONE-D-25-17586R2

PLOS ONE

Dear Dr. Flores-Kanter,

I'm pleased to inform you that your manuscript has been deemed suitable for publication in PLOS ONE. Congratulations! Your manuscript is now being handed over to our production team.

Kind regards,

on behalf of

Dr. Diogo Manuel Teixeira Monteiro

Academic Editor

PLOS ONE